# Normative Data of Objectively Measured Physical Activity and Sedentary Time in Community—Dwelling Older Japanese

**DOI:** 10.3390/ijerph18073577

**Published:** 2021-03-30

**Authors:** Harukaze Yatsugi, Tao Chen, Si Chen, Kenji Narazaki, Sho Nagayoshi, Shuzo Kumagai, Hiro Kishimoto

**Affiliations:** 1Department of Behavior and Health Sciences, Graduate School of Human-Environment Studies, Kyushu University, 744 Motooka Nishi-ku, Fukuoka City, Fukuoka 819-0395, Japan; haru19920424@gmail.com (H.Y.); sho.nagayoshi@omron.com (S.N.); 2Sports and Health Research Center, Department of Physical Education, Tongji University, Shanghai 200092, China; chentwhy@tongji.edu.cn; 3School of Nursing and Rehabilitation, Cheeloo College of Medicine, Shandong University, 44 West Wenhua Road, Jinan 250012, China; chensi2jap@yahoo.co.jp; 4Faculty of Socio-Environmental Studies, Fukuoka Institute of Technology, 3-30-1 Wajiro-higashi, Higashi-ku, Fukuoka City, Fukuoka 811-0295, Japan; narazaki@fit.ac.jp; 5Development Center, Clinical Development Department, Technology Development HQ, Omron Healthcare Co., Ltd., 53 Kunotsubo, Terado-cho, Muko, Kyoto 617-0002, Japan; 6Institute of Covergence Bio-Health, Dong-A University, B08-0302, 37 Nakdong-daero 550 Beon-gil, Saha-gu, Busan 49315, Korea; kumagai.shuzo.296@m.kyushu-u.ac.jp; 7Faculty of Arts and Science, IC15, Kyushu University, 744 Motooka Nishi-ku, Fukuoka City, Fukuoka 819-0395, Japan

**Keywords:** physical activity, sedentary time, tri-axial accelerometer, older adult, Japanese

## Abstract

Background: The amounts of moderate-to-vigorous-intensity physical activity (MVPA), light-intensity physical activity (LPA), and sedentary time (ST) by sex, age, and body mass index (BMI) in older Japanese adults have not been known. We conducted this study to determine the actual physical activity (PA) and ST in this population. Subjects and Methods: A total of 3998 community-dwelling Japanese adults aged ≥65 years were investigated. Their levels of PA and ST and number of steps taken daily were assessed for seven consecutive days by a tri-axial accelerometer. Normative values of daily PA and ST were analyzed by age and BMI groups in the men and the women and are presented as mean, median, or decile. Results: The subjects generally adhered to the PA guideline, i.e., ≥10 metabolic equivalents (METs)·hour MVPA per week. Older age was associated with lower adherence to the PA guideline. Conclusions: Normative values (mean, median, or decile) were yielded for MVPA, LPA, and ST based on accelerometer readings in a large sample of older community-dwelling Japanese adults. One-half of the subjects’ waking time was spent being sedentary, and >70% of the subjects met the current PA guideline by engaging in MVPA.

## 1. Introduction

Physical activity (PA) is increasingly recognized as a factor that can decrease the risks of obesity, type 2 diabetes, cardiovascular disease, osteoporosis, and mortality [1,2,3,4,5]. The Japanese Physical Activity Guideline for Health 2013 was thus designed to help Japan’s population increase their PA in daily life to improve and maintain their health [6]. This PA Guideline recommends that adults (aged 18–64 years) should engage in moderate-to-vigorous-intensity physical activity (MVPA) for 4 metabolic equivalents (METs)·hours/week (h/wk), and that older adults (≥65 years) should engage in PA regardless of its intensity for 10 METs·h/wk. At this juncture, intensity of PA and sedentary time (ST) were not mentioned in Japan’s PA guideline of older adults.

However, Japan’s PA guideline was standardized based on the findings of a meta-analysis that included four studies of western populations and using self-reported PA data [7,8,9,10]. There appears to be no data to describe such prevalence at a Japanese population level.

An accelerometer allows objective and accurate measurements of PA at the individual and population levels, and its use reduces the potential assessment bias that is inherent to self-reported measures especially about older adults. There are few objectively measured PA data available for a large-scale Japanese older population (or, the existing data are limited to a single area or a limited sample size, which may limit the generalization of the findings) [11,12,13]. A pooled analysis that combines cohorts from multiple areas could improve our understanding of the levels of PA in older Japanese and thus provide normative PA data of community-dwelling older adults. The design of the present study thus covers multiple areas and a large sample that can provide precise PA and ST data in older Japanese adults, obtained with the use of a tri-axial accelerometer.

## 2. Subjects and Methods

### 2.1. Study Subjects

The study population was drawn from four municipalities conducted in Fukuoka Prefecture, one of Japan’s 47 prefectures. In these municipalities, residents aged ≥65 years old who participated in fitness assessments were included in this study. Individuals with long-term care needs were excluded from the pre-screening process. The eligible subjects were not certified as requiring long-term care by Japan’s long-term care insurance system.

A total of 5635 subjects were eligible for the study [11,12,14,15], and 3998 of the subjects (1167 men and 2327 women) who provided valid accelerometer data (i.e., ≥10 h for ≥4 days of wear time) were enrolled. These data were acquired in 2009 (Spring–Summer) [12], in 2009 (Summer–Autumn) [14], in 2011 (Spring) [11], and in 2017 (Autumn) [15]. This study was approved by the Institutional Review Board of Kyushu University, Japan. All subjects provided written informed consent to participate.

### 2.2. Physical Activity and Sedentary Behavior Measures

Each subject’s PA and ST were objectively measured by a tri-axial accelerometer (Active Style Pro HJA-350IT, Omron Healthcare, Kyoto, Japan). The subjects were instructed to wear the accelerometer on the right or left side of their waist for 7 consecutive days and remove it only before going to bed or participating in water activities. Data were recorded in 1-min epochs. The accuracy of the intensity estimated by the Active Style Pro has been validated with the Douglas bag method [16]. Additionally, the use of a tri-axial accelerometer to assess PA and sedentary time allowed for a more accurate estimate of activity intensity than a conventional uni-axial accelerometer [16]. Technical specification and data acquisition system for the Active style Pro have been previously reported. [16,17]. Non-wear time was defined as ≥60 consecutive min of no activity, i.e., estimated activity intensity <1.0 metabolic equivalents (METs) with an allowance for 2 min of activities with an intensity up to 1.0 METs [11,18,19]. The data of only the subjects with ≥4 valid wear days (≥10 h of wear time/day) were included in the analysis [20].

The cut-off values used to define the amounts of time spent in ST, light-intensity physical activity (LPA), and MVPA were as follows: ≤1.5 METs for ST, 1.6–2.9 METs for LPA, and ≥3.0 METs for MVPA. As noted above, the Japanese Physical Activity Guideline for Health 2013 states that older adults (≥65 years) should engage in PA (regardless of its intensity) for 10 METs·hr/wk [6], and the goals of the 2nd edition of Healthy Japan 21 for older adults are 7000 daily steps for men and 6000 daily steps for women [21]. This study determined how many of the subjects performed MVPA at 10 METs·hr/wk.

### 2.3. Other Measures

Each subject’s body weight (kg) and body height (m) were measured using standard protocols with the subject in light clothing and without shoes. The subjects’ body mass index (BMI) values were calculated as weight in kilograms divided by height in meters squared. We then categorized the BMI values into <18.5, 18.5–24.9, 25.0–29.9, and ≥30.0 kg/m^2^ groups. We categorized the subjects’ ages into 65–69, 70–74, 75–79, 80–84, and ≥85 years as age groups.

### 2.4. Statistical Analyses

The time spent at each defined intensity of PA or sedentary was summed over the valid days for each subject, and the daily averages were then calculated. The proportions of subjects in each of the categories were also calculated and are presented for the age, sex, and BMI groups. Descriptive data were summarized as means ± standard deviation (SD), median, and/or each ten percentile (decile) for continuous variables and as the frequency (percentage) for categorical variables.

We conducted an analysis of variance (ANOVA) to identify differences in the proportion of subjects meeting the current Japan PA guideline (i.e., PA regardless of its intensity for 10 METs·hr/wk) among the sex, age, and BMI groups. All statistical analyses were conducted using SAS ver. 9.4 (SAS, Cary, NC, USA). The significance level was set at two-sided α = 0.05.

## 3. Results

### 3.1. Population

The median age of all 3,998 subjects was 72 years, and the median BMI was 22.9 kg/m^2^. Women comprised 58.2% (n = 2327) of the subjects; men accounted for 41.8% (n = 1671). The median steps for the men and women were 5701 and 5038, respectively. The median accelerometer wearing time was 803 min/day in the men and 846 min/day in the women. Men spent 39.6 min/day in MVPA, 283.6 min/day in LPA, and 478.0 min/day in ST. Compared to the men, the women spent more time engaging in MVPA (43.2 min/day) and LPA (373.1 min/day) and less ST (417.9 min/day).

### 3.2. PA and ST

Table 1 summarizes the distribution of steps, accelerometer wearing time, and time spent in MVPA, LPA, and ST according to age categories in the men. The mean or median values of steps, accelerometer wearing time, MVPA, and LPA were each lower with increasing age, whereas the ST values were greater in the higher age groups. According to the mean values of each PA variable, the younger age groups exhibited significantly high MVPA (*p* < 0.001) and LPA (*p* < 0.001) and significantly low ST (*p* < 0.001) compared to the older age groups. Similar findings were observed in the women (Table 2).

We next investigated the distributions of values according to BMI categories in the men (Table 3) and women (Table 4). In both groups, the subjects in the second BMI group (18.5–24.9) values showed the highest numbers of steps taken and the longest MVPA durations. According to the mean values of each PA variable, the lower BMI groups among the men showed significantly high LPA (*p* < 0.0001) and significantly low ST (*p* = 0.0088). There were also significant differences (*p* < 0.05) between the 18.5–24.9 group and <18.5 group or 25.0–29.9 group in MVPA, and 25.0–29.9 group or ≥30.0 group in LPA (Table 3). In the group of women, the lower BMI groups demonstrated significantly high MVPA (*p* = 0.0009), significantly high LPA (*p* < 0.0001), and significantly low ST (*p* < 0.0001). There were also significant differences (*p* < 0.05) between the 18.5–24.9 group and 25.0–29.9 group in MVPA, <18.5 group or 25.0–29.9 group or ≥30.0 group in LPA, and 25.0–29.9 group or ≥30.0 group in ST (Table 4).

### 3.3. PA Guideline Adherence

As shown in Figure 1, 71.0% of the subjects adhered to the PA guideline (defined as ≥10 METs·h/wk) by engaging in MVPA only. With regard to sex-specific prevalence, more women adhered to the PA guideline (73.1%) compared to men (68.1%). The adherence to the PA guideline decreased with age in both men and women (*p* < 0.0001). As shown in Figure 2, the adherence to the PA guideline also decreased with increasing BMI starting from the second BMI group (72.7%) through the higher-BMI groups (highest group: 65.6%) and both sexes (*p* = 0.0028).

### 3.4. Associations

For the entire population, there was a significant positive correlation between MVPA and LPA (r = 0.30) and a significant negative correlation between ST and MVPA (r = −0.42) or LPA (r = −0.60) (both *p* < 0.001).

## 4. Discussion

The results of our analyses demonstrated that among nearly 4000 Japanese adults aged ≥65, the median daily steps taken by men was 5701 and that taken by women was 5038. The higher number of daily steps was more likely among men, the younger age groups, and the second BMI group. Overall, the subjects spent approx. 5.0% of their waking time in MVPA, 41.1% in LPA, and 53.9% in ST. In addition, 71.0% of the subjects reached the recommended criteria for the Japanese PA guideline in 2013 when it was calculated only with MVPA.

### 4.1. PA and ST

The Nakanojo Study of Japanese subjects reported that all of the men and women exceeded ~8000 and 6900 steps/day, respectively [22]. This finding is close to the goal of the 2nd edition of Healthy Japan 21 for older adults, i.e., 7000 daily steps for men and 6000 daily steps for women [21]. The reasons for the inconsistencies in the numbers of steps between the Nakanojo Study and our present observations might be due to the different sample sizes (n = 175 and n = 3998) and differing measurement devices, i.e., a uni-axial accelerometer (modified Kenz Lifecorder, Suzuken Co., Nagoya, Japan) in the previous study and a tri-axial accelerometer (Active style Pro HJA-350IT) in the present study.

We observed a sex difference regarding the times spent in MVPA, LPA, and ST in this population, which agrees with previous investigations conducted in Japan [12,13] but not western studies [23]. It is possible that PA patterns differ between men and women depending on customs and cultures. Regarding BMI values, in the present study, the MVPA in men’s lowest BMI group was as much as that in the men’s highest BMI group, but the LPA and ST were quite different between these BMI groups. With regard to the differences in PA among the present age and BMI groups, our finding that the MVPA and LPA values are greater in subjects with lower BMIs and in younger individuals is common around the world [23].

### 4.2. PA Guideline Adherence

An American guideline recommends that adults engage in ≥150 min/week of moderate-intensity physical activity (MPA), or 75 min/week of vigorous-intensity physical activity (VPA), or a combination of MVPA. However, a study that examined the adherence to this guideline revealed that <10% of older American adults met the guideline [24]. The index of 10 METs·hr is equivalent to 4 METs (MVPA) 2.5 h (150 min) of PA. In our present population, 71.0% of the subjects adhered to the PA guideline (defined as ≥10 METs·h/wk) with only their MVPA, and this result demonstrates that the proportion of older Japanese who meet the guideline is quite high, although the subjects in the oldest age group showed lower adherence to the PA guideline than the younger age groups. In addition, our subjects in the second BMI group exhibited higher adherence to the PA guideline than the higher-BMI groups.

The American College of Sports Medicine reported that regular exercise can both minimize the physiological effects of an otherwise sedentary lifestyle and increase an individual’s active life expectancy by limiting the development and progression of chronic disease and disabling conditions [25]. In light of that report, a higher level of PA may be necessary for older people irrespective of sex, age, and BMI level.

Regarding the Japanese Physical Activity Guideline for Health 2013, a larger proportion of women than men adhered to the PA guideline (≥10 METs·h/wk) only by MVPA in this study. In the youngest group (aged 65–69 years) in both sexes, 85.0% of the subjects adhered to the PA guideline. In contrast, 25.5% of the oldest group (≥85 years) adhered to this guideline. Older adults’ circumstances are likely to have undergone changes such as retirement and age-related illnesses [26]. A longitudinal study showed that the level of PA decreased rapidly among community-dwelling adults aged ≥60 [27]. In addition to changing circumstances, the amount of MVPA might decrease with a decline in older adults’ quality of life [28]. These findings highlight that the recommended values of PA and MVPA should take the ages of older adults into account.

### 4.3. Study Strengths and Limitations

The strengths of this study include the relatively large Japanese population (over 3000 people) aged ≥65 years and the use of a tri-axial accelerometer to assess PA and ST, which provides a more accurate estimate of activity intensity than questionnaires. Our findings comprise the first detailed description of physical activity and sedentary time with regard to age, sex, and BMI in the southwest region of Japan.

Some limitations should be considered when interpreting our findings. (1) The recognized limitations of accelerometers include their inability to detect some types of PA (e.g., water activities and cycling) and distinguish between postures. In addition, the Active Style Pro HJA-350IT might underestimate MVPA among older adults [29]. (2) Almost all of our subjects were without long-term care and assistance; they might be relatively active and motivated compared to the general population. (3) There is a possibility that the subjects’ living environments affected their PA levels; the PA may also differ among the various sizes of neighborhood spaces. (4) The median accelerometer wearing times in men and women were 803.0 min/day and 846 min/day, respectively. Since the minutes of PA and ST depend on the accelerometer wearing time, it is important to consider the wearing time. A recent Japanese study reported that the mean wear time was 854.9 min/day in men and 898.6 min/day in women [13]. This result agrees with ours with respect to the sex difference; i.e., men wore the accelerometer less than women did. Judging from this result, the wearing time may reflect the wearer’s lifestyle/behavior. Moreover, the difference in wearing times (shorter wear times were observed herein compared to the previous study) might be due in part to the different regions from which the subjects were drawn. In the previous study [13], two-thirds of the subjects lived in the Tokyo Metropolitan area, which is the most urban area in Japan. It is possible that the region (e.g., urban vs. suburban and rural) influences individual lifestyles.

## 5. Conclusions

Our analyses revealed that about one-half of the waking time of this Japanese population of older adults was spent being sedentary. Moreover, the percentage of the subjects’ time spent in MVPA was low regardless of sex, age, and BMI status. All of the subjects met the current physical activity guideline in Japan, and >70% of them met this guideline by engaging in only MVPA.

## Figures and Tables

**Figure 1 ijerph-18-03577-f001:**
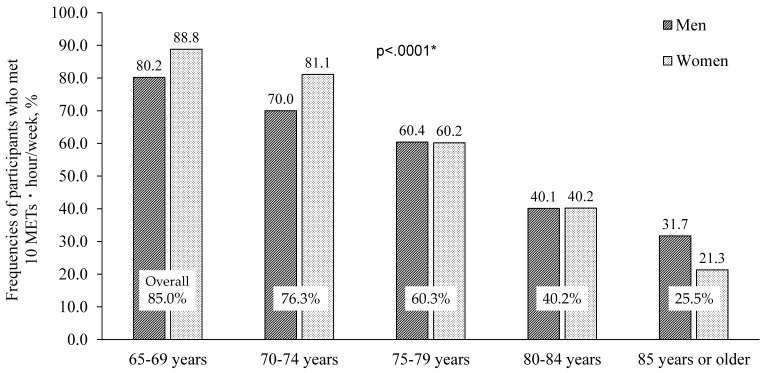
Percentage of participants who met 10 metabolic equivalents (METs)·hour/week guideline only considering MVPA by sex and age.

**Figure 2 ijerph-18-03577-f002:**
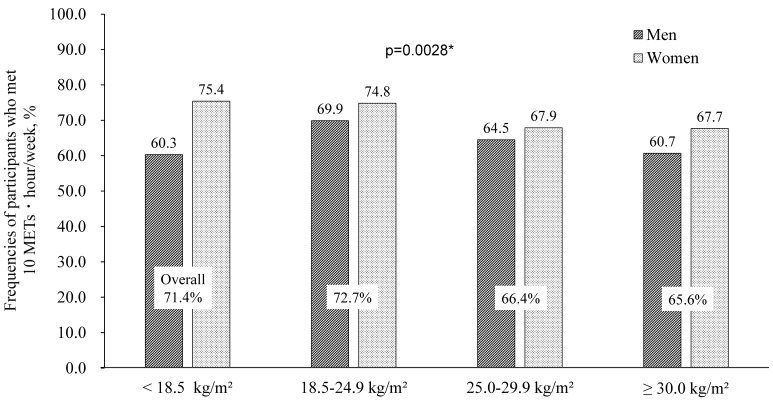
Percentage of participants who met 10 METs·hour/week guideline only considering MVPA by sex and BMI.

**Table 1 ijerph-18-03577-t001:** Tri-axial accelerometer-determined PA and ST by age groups in the men (n = 1671).

	Overall (n = 1671)	Age Categories, yrs					
		65–69 (n = 625)	70–74 (n = 526)	75–79 (n = 323)	80–84 (n = 137)	≥85 (n = 60)	*p* for Trend
**Steps:**							
Mean (SD)	5701 (3398)	6542 (3681)	5701 (2990)	5250 (3127)	4041 (2965)	3156 (2963)	
Median	5186	5943	5454	4672	3295	2519	
**AWT, min/day:**							
Mean (SD)	811.8 (94.2)	818.3 (98.5)	809.4 (87.3)	807.7 (92.0)	809.6 (101.7)	793.7 (98.6)	
Median	802.8	814.4	800.4	798.1	802.7	778.0	
**MVPA, min/day:**							
Mean (SD)	46.1 (34.0)	55.1 (36.4)	47.2 (31.3)	39.8 (31.5)	26.8 (25.2)	21.5 (23.3)	<0.0001
Median	39.6	48.8	40.6	32.3	18.0	13.9	
decile p10	9.5	15.9	12.9	6.5	4.4	2.0	
decile p20	16.9	25.7	19.4	13.1	7.3	3.3	
decile p30	24.3	34.6	25.1	19.8	9.7	7.2	
decile p40	32.5	41.6	34.4	26.2	13.5	10.9	
decile p60	48.1	57.7	49.4	39.5	25.9	16.5	
decile p70	57.9	66.4	59.7	50.3	35.8	25.9	
decile p80	70.3	78.6	74.1	59.5	43.8	36.8	
decile p90	90.0	97.8	90.3	83.6	59.2	45.0	
**LPA, min/day:**							<0.0001
Mean (SD)	286.4 (87.8)	298.6 (87.7)	285.8 (86.5)	281.4 (81.2)	267.6 (96.3)	235.1 (88.0)	
Median	283.6	294.6	285.7	279.5	257.4	233.1	
decile p10	175.3	188.7	177.3	181.4	154.0	126.6	
decile p20	210.5	224.5	210.5	205.3	182.3	150.9	
decile p30	236.5	248.8	237.9	230.8	215.2	172.9	
decile p40	260.6	273.4	266.4	253.8	233.6	200.5	
decile p60	306.3	314.3	307.3	300.9	277.1	262.9	
decile p70	330.4	341.2	329.6	323.1	301.7	282.5	
decile p80	359.3	367.2	360.9	354.4	347.1	326.7	
decile p90	398.8	414.8	393.6	386.0	414.8	342.7	
**ST, min/day:**							
Mean (SD)	479.3 (122.5)	464.5 (125.4)	476.5 (120.2)	486.5 (109.2)	515.2 (128.3)	537.0 (132.4)	<0.0001
Median	478.0	469.6	471.0	484.4	513.8	538.9	
decile p10	321.4	301.5	323.3	346.4	371.3	369.7	
decile p20	379.4	357.2	378.8	396.4	432.6	421.5	
decile p30	419.4	404.0	417.0	433.3	472.3	469.4	
decile p40	450.3	435.9	449.2	459.6	491.3	497.2	
decile p60	509.9	499.6	501.7	513.7	541.1	578.3	
decile p70	540.3	530.1	535.0	542.8	573.0	613.9	
decile p80	575.4	563.0	570.1	577.7	603.4	644.5	
decile p90	621.1	599.5	621.3	617.2	639.7	712.2	

AWT: accelerometer wearing time, LPA: light physical activity, MVPA: moderate-to-vigorous physical activity, ST: sedentary time.

**Table 2 ijerph-18-03577-t002:** Tri-axial accelerometer-determined PA and ST by age groups in the women (n = 2327).

	Overall (n = 2327)	Age categories, yrs					
		65–69 (n = 813)	70–74 (n = 713)	75–79 (n = 483)	80–84 (n = 229)	≥85 (n = 89)	*p* for Trend
**Steps:**							
Mean (SD)	5037 (2786)	6104 (2901)	5331 (2498)	4230 (2323)	3184 (2179)	2105 (1627)	
Median	4596	5539	5020	3885	2820	1696	
**AWT, min/day:**							
Mean (SD)	849.9 (97.1)	864.9 (91.6)	851.3 (92.4)	839.0 (105.1)	829.6 (100.9)	813.6 (103.7)	
Median	845.8	861.7	851.7	830.5	819.3	809.2	
**MVPA, min/day:**							
Mean (SD)	50.4 (35.7)	65.5 (38.6)	52.3 (32.3)	39.6 (29.1)	26.5 (22.2)	18.5 (17.3)	<0.0001
Median	43.2	57.4	45.9	33.0	19.9	13.0	
decile p10	11.3	24.1	18.9	8.0	4.0	2.9	
decile p20	20.3	33.5	26.4	14.0	7.8	3.8	
decile p30	28.6	41.1	33.1	19.9	12.3	7.3	
decile p40	35.8	49.4	39.1	26.6	16.0	9.3	
decile p60	51.8	65.9	53.0	40.9	26.5	17.2	
decile p70	61.0	79.7	61.0	49.8	33.7	22.4	
decile p80	76.9	96.0	75.8	60.9	41.3	32.0	
decile p90	99.2	114.3	95.7	82.6	55.0	44.7	
**LPA, min/day:**							
Mean (SD)	374.4 (85.7)	391.4 (80.9)	380.1 (81.0)	364.5 (87.2)	343.8 (88.6)	306.5 (89.4)	<0.0001
Median	373.1	390.6	377.3	361.3	345.8	300.0	
decile p10	266.1	287.8	280.8	255.3	235.1	186.7	
decile p20	300.1	323.9	315.5	285.8	260.0	238.5	
decile p30	329.6	349.9	334.9	319.6	293.6	266.7	
decile p40	352.0	371.3	358.3	339.6	316.8	271.3	
decile p60	395.0	411.0	402.0	382.4	367.3	325.9	
decile p70	419.6	434.5	422.7	407.9	385.0	340.5	
decile p80	448.1	459.6	449.8	442.5	411.9	386.6	
decile p90	486.6	498.5	481.1	484.3	470.4	446.5	
**ST, min/day:**							
Mean (SD)	425.1 (112.3)	408.0 (107.4)	418.9 (102.1)	435.0 (122.4)	459.3 (119.5)	488.6 (112.9)	<0.0001
Median	417.9	405.4	409.4	424.2	450.1	487.3	
decile p10	289.4	279.5	293.9	291.0	309.1	337.3	
decile p20	333.4	314.4	337.0	338.8	364.6	399.8	
decile p30	364.6	350.0	361.0	378.1	401.7	437.3	
decile p40	392.1	376.8	385.0	400.3	429.0	473.4	
decile p60	444.4	430.8	437.3	448.8	485.6	520.0	
decile p70	476.6	456.8	467.4	483.0	511.4	540.7	
decile p80	513.1	493.4	506.4	526.4	547.0	582.9	
decile p90	565.4	544.4	550.3	572.5	611.8	605.9	

AWT: accelerometer wearing time, LPA: light physical activity, MVPA: moderate-to-vigorous physical activity, ST: sedentary time.

**Table 3 ijerph-18-03577-t003:** Tri-axial accelerometer-determined PA and ST by BMI groups in the men (n = 1671).

	Overall (n = 1671)	BMI categories, kg/m^2^				
		<18.5 (n = 73)	18.5–24.9 (n = 1187)	25.0–29.9 (n = 383)	≥30.0 (n = 28)	*p* for Trend
**Steps**						
Mean (SD)	5701 (3398)	4976 (2995)	5888 (3323)	5324 (3697)	4820 (2503)	
Median	5186	4699	5414	4692	4716	
**AWT, min/day:**						
Mean (SD)	811.8 (94.2)	810.8 (96.5)	813.8 (94.2)	807.0 (93.2)	799.1 (105.0)	
Median	802.8	807.7	803.4	799.6	773.4	
**MVPA, min/day:**						
Mean (SD)	46.1 (34.0)	36.1 (28.0)	48.3 (34.3)	42.1 (34.0)	35.4 (22.6)	0.06
Median	39.6	32.4	42.3	36.1	30.8	
decile p10	9.5	6.5	10.0	9.0	7.0	
decile p20	16.9	11.5	17.8	15.6	13.1	
decile p30	24.3	14.8	25.7	22.1	17.5	
decile p40	32.5	25.7	33.8	29.9	25.4	
decile p60	48.1	38.5	51.6	41.8	43.4	
decile p70	57.9	43.0	61.4	50.1	51.2	
decile p80	70.3	49.4	74.1	61.4	57.0	
decilep90	90.0	86.4	92.4	80.0	61.5	
**LPA, min/day:**						
Mean (SD)	286.4 (87.8)	294.2 (104.1)	291.4 (86.0)	272.4 (88.2)	247.0 (85.5)	<0.0001
Median	283.6	282.2	289.3	269.6	234.2	
decile p10	175.3	172.5	181.4	162.6	114.8	
decile p20	210.5	228.8	215.2	198.3	177.6	
decile p30	236.5	242.3	243.2	223.5	210.0	
decile p40	260.6	254.4	267.0	247.3	215.8	
decile p60	306.3	302.3	312.6	291.0	268.3	
decile p70	330.4	343.0	334.7	309.8	285.0	
decile p80	359.3	372.3	362.6	349.3	330.9	
decile p90	398.8	445.5	403.0	383.8	378.4	
**ST, min/day:**						
Mean (SD)	479.3 (122.5)	480.4 (134.5)	474.1 (121.6)	492.5 (119.6)	516.8 (148.0)	0.0088
Median	478.0	488.6	473.3	493.6	493.2	
decile p10	321.4	332.9	319.7	339.0	363.3	
decile p20	379.4	376.8	371.8	404.3	391.0	
decile p30	419.4	415.0	412.4	439.8	428.8	
decile p40	450.3	452.5	445.4	470.6	449.1	
decile p60	509.9	523.4	503.6	525.3	562.5	
decile p70	540.3	547.6	535.4	550.5	586.5	
decile p80	575.4	590.6	571.6	577.1	609.5	
decile p90	621.1	621.3	621.8	615.6	654.3	

AWT: accelerometer wearing time, LPA: light physical activity, MVPA: moderate-to-vigorous physical activity, ST: sedentary time.

**Table 4 ijerph-18-03577-t004:** Tri-axial accelerometer-determined PA and ST by BMI groups in the women (n = 2327).

	Overall (n = 2327)	BMI Categories, kg/m^2^				
		<18.5 (n = 207)	18.5–24.9 (n = 1547)	25.0–29.9 (n = 505)	≥30.0 (n = 68)	*p* for Trend
**Steps**						
Mean (SD)	5037 (2786)	4964 (2867)	5173 (2781)	4736 (2759)	4424 (2630)	
Median	4596	4716	4774	4402	3861	
**AWT, min/day:**						
Mean (SD)	849.9 (97.1)	863.7 (97.4)	848.1 (94.1)	849.2 (104.8)	854.9 (102.4)	
Median	845.8	859.6	845.5	839.8	841.1	
**MVPA, min/day**						
Mean (SD)	50.4 (35.7)	51.6 (37.5)	52.0 (35.5)	46.4 (36.0)	40.9 (28.9)	0.0009
Median	43.2	44.9	45.0	38.8	35.7	
decile p10	11.3	10.9	13.2	8.3	8.4	
decile p20	20.3	20.8	21.7	17.1	14.6	
decile p30	28.6	29.8	30.3	24.1	23.0	
decile p40	35.8	36.0	37.2	33.5	32.3	
decile p60	51.8	53.9	53.8	46.6	39.0	
decile p70	61.0	60.8	63.3	56.6	49.9	
decile p80	76.9	75.0	80.0	68.3	59.7	
decilep90	99.2	100.6	101.0	93.1	84.2	
**LPA, min/day**						
Mean (SD)	374.4 (85.7)	396.3 (87.9)	378.3 (84.4)	357.3 (83.9)	346.9 (94.8)	<0.0001
Median	373.1	395.8	377.1	354.4	340.2	
decile p10	266.1	275.0	270.6	253.5	230.6	
decile p20	300.1	327.0	307.8	284.0	266.9	
decile p30	329.6	352.5	333.7	308.8	281.3	
decile p40	352.0	371.3	356.3	333.6	311.5	
decile p60	395.0	416.5	399.8	378.2	376.3	
decile p70	419.6	444.6	423.4	396.6	389.8	
decile p80	448.1	473.6	448.7	430.9	431.3	
decilep90	486.6	509.0	487.6	468.3	466.4	
**ST, min/day**						
Mean (SD)	425.1 (112.3)	415.8 (104.6)	417.8 (109.0)	445.5 (120.0)	467.1 (123.1)	<0.0001
Median	417.9	408.5	412.3	438.3	463.3	
decile p10	289.4	291.2	285.4	301.6	365.0	
decile p20	333.4	329.3	328.4	348.1	409.9	
decile p30	364.6	355.6	359.9	380.8	436.9	
decile p40	392.1	382.3	386.0	405.2	463.3	
decile p60	444.4	431.1	438.6	466.9	507.0	
decile p70	476.6	456.1	468.0	498.9	539.9	
decile p80	513.1	496.8	505.1	541.1	566.9	
decilep90	565.4	558.9	554.1	590.9	630.6	

AWT: accelerometer wearing time, LPA: light physical activity, MVPA: moderate-to-vigorous physical activity, ST: sedentary time.

## Data Availability

The statistical analyses were carried out using the computer resource offered under the category of General Projects by the Research Institute for Information Technology, Kyushu University, Japan.

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
