# Peer review of "Normative Data of Objectively Measured Physical Activity and Sedentary Time in Community—Dwelling Older Japanese"

_ijerph, 2021, doi:10.3390/ijerph18073577_

Round 1

Reviewer 1 Report

In this study, the different physical activity levels (PAL) of the elderly in Japan were measured by the three-axis accelerometers according to age, gender, and BMI. In general, the data presented here show some value, but its experimental design is incomplete, which greatly reduces the scientific value of this article.

1 This research first ignores some key factors in the experimental design
1 ) The difference in PAL is not related to age, gender, and BMI index, but also affected by other characteristics of the individual, such as personal income, education level, occupation before retirement, etc.;
2 ) The difference in  PAl may also come from the difference in neighborhood space. The accessibility and quality of surrounding fitness facilities will also affect the fitness behavior of community residents
3 )There may be endogenous problems with body mass index and PAL. Do people with low BMI keep the low value by exercise, or do they exercise because of high BMI?

2 The relationship between the measurement of PAL in this study and other related studies has not been clearly explained, such as TEE BMR.

The PALs used in the study, LPA and MVPA, focus only on the duration of exercise, which is different from the division in the WHO organization and Healthy People, which refers to the intensity of exercise. 

3 The research gap, discussion, and significance of this research all need more elaboration.

Reviewer 2 Report

Dear authors,

dear editos,

thank you for giving me the opportunity to review this paper. From my point of view, it adequatly introduce to the topic and a clear study purpose is made. Material and methods are well described. The result secion and the discussion are well written. I have only some minor aspects which needs to be adressed.

l.28/ l.42 Define MET (metabolic equivalent of task)

l.29: „ normative values“ – please provide more information for unexperienced readers like „normative values compared with the Western population etc., as you state in your introduction.

l.63: „old who participated in fitness assessments were included in the study.“ What does „ fitness assessments“ exactly mean? Might this lead to bias?

l.67: please provide some of the inclusion crieteria in short or give further information. It is not suffiecient to only state 4 references.

l.161: This section may be divided by subheadings. It should provide a concise and precise 161 description of the experimental results, their interpretation, as well as the experimental 162 conclusions that can be drawn. àerase this part, which is a standard sentence of the template.

Figure 1/2: please provide an asterixs for possible significant differences (*)

Please provide the year/ years of data acquistion. Please provide the season of data acquistion. Maybe in winter physical activity might decrease.

Round 2

Reviewer 1 Report

The authors had provide responses according to the reviewing points, which added additional information onto this paper, which may be helpful to readers 

However,  the introdution still need more revision

1 introduction of  traditional  measurement of PAL , why  tri-axial accelerometer is accomended

2 what factors  affect the PAL level of  older Japanese 

3 what is the result  of unclear measurement of PAL in terms of  healthcare policy
